# The Efficacy and Safety of Direct Oral Anticoagulants versus Standard of Care in Patients without an Indication of Anti-Coagulants after Transcatheter Aortic Valve Replacement: A Meta-Analysis of Randomized Controlled Trials

**DOI:** 10.3390/jcm11226781

**Published:** 2022-11-16

**Authors:** Mohamed Abuelazm, Basel Abdelazeem, Basant E. Katamesh, Mohamed Gamal, Lakshmi Venkata Simhachalam Kutikuppala, Babikir Kheiri, James Robert Brašić, Timir K. Paul

**Affiliations:** 1Faculty of Medicine, Tanta University, Tanta 31527, Egypt; 2Department of Internal Medicine, McLaren Health Care, Flint, MI 48532, USA; 3Department of Internal Medicine, Michigan State University, East Lansing, MI 48823, USA; 4Faculty of Modern Medicine, Dr. N.T.R. University of Health Sciences, Vijayawada 520008, Andhra Pradesh, India; 5Electrophysiology and Arrhythmias Service, Division of Cardiology, Department of Medicine, University of California, San Francisco, CA 94117, USA; 6Section of High-Resolution Brain Positron Emission Tomography Imaging, Division of Nuclear Medicine and Molecular Imaging, The Russell H. Morgan Department of Radiology and Radiological Science, The Johns Hopkins University School of Medicine, Baltimore, MD 21287, USA; 7Department of Clinical Medical Education, The University of Tennessee Health Sciences Center at Nashville, Nashville, TN 38163, USA

**Keywords:** aortic stenosis, apixaban, atrial fibrillation, cerebral hemorrhage, confidence interval, edoxaban, flow chart, rivaroxaban, thrombosis, valvular heart diseases

## Abstract

Transcatheter aortic valve replacement (TAVR) is now considered the mainstay of aortic stenosis management; however, the optimal antithrombotic therapy in patent without indications for an oral anticoagulant (OAC) is yet to be identified. Therefore, we conducted a systematic review and meta-analysis to evaluate the efficacy and safety of direct oral anticoagulant (DOAC) treatment versus the standard of care in patients without indications of OACs after TAVR. We synthesized randomized controlled trials (RCTs) from Web of Science, SCOPUS, EMBASE, PubMed, and Cochrane until 18 August 2022. We used the risk ratio (RR) for dichotomous outcomes with the corresponding 95% confidence interval (CI). We registered our protocol in PROSPERO with ID: CRD42022357027. Three RCTs with 2922 patients were identified. DOACs were significantly associated with higher incidence of all-cause mortality (RR: 1.68 with 95% CI [1.22, 2.30], *p* = 0.001), mortality due to non-cardiovascular causes (RR: 2.34 with 95% CI [1.36, 4.02], *p* = 0.002), and the composite outcome of death, myocardial infarction, or stroke (RR: 1.41 with 95% CI [1.13, 1.76], *p* = 0.002). However, DOACs were associated with decreased incidence of reduced leaflet motion (RLM) (RR: 0.19 with 95% CI [0.09, 0.41], *p* = 0.0001) and hypoattenuated leaflet thickening (HALT) (RR: 0.50 with 95% CI [0.36, 0.70], *p* = 0.0001). DOACs were effective to reduce RLM and HALT; however, the clinical effect of this is still controversial. DOACs were associated with worse efficacy and safety outcomes, including all-cause mortality. Further RCTs investigating the optimal antithrombotic regimen after TAVR.

## 1. Introduction

Transcatheter aortic valve replacement (TAVR) is an established approach in the management of symptomatic severe aortic stenosis. To avoid thrombotic complications following TAVR, antithrombotic medication is necessary, despite its increased chance of bleeding events [1]. TAVR patients are usually elderly and have other comorbid conditions, such as atrial fibrillation (AF), which may increase the risk of thrombosis. Thus, the complex mechanism underlying thrombotic events following TAVR points to the necessity of effective antithrombotic therapy, which may include antiplatelet and anticoagulant therapy [2].

Patients with bioprosthetic aortic valve implanted by transcatheter mode who need anticoagulation for other indications are advised to take lifelong oral anticoagulation (OAC). The direct oral anticoagulants (DOACs) especially the Factor Xa inhibitors are easy to adopt and were linked to comparable stroke outcomes and considerably decreased bleeding cerebral hemorrhage, and mortality risks for one year when compared with the other drug classes such as the vitamin K antagonists [3].

The latest randomized control trials (RCTs) showed that single antiplatelet therapy (SAPT) has reduced bleeding rates with an equivalent risk of ischemic outcomes, compared with dual antiplatelet therapy (DAPT) in patients without an indication of OAC [4,5,6,7]. Therefore, the recent ESC/EACTS 2021 and American College of Cardiology/American Heart Association (ACC/AHA) 2020 guidelines have recommended SAPT for patients without an indication of OAC [8,9].

Recently, the decreased incidence of reduced leaflet motion (RLM) and hypoattenuated leaflet thickening (HALT) using the DOACs, such as the most commonly used factor Xa inhibitors, appears promising in improving valve durability and hemodynamics [6,7,10,11,12]. However, DOACs have been associated with an increased risk of mortality and adverse cardiac events in patients without an indication for OAC after TAVR. Therefore, we aim to evaluate the safety and efficacy of DOACs in patients without an indication of OAC after TAVR.

## 2. Materials and Methods

### 2.1. Protocol Registration

Our systematic review and meta-analysis was prospectively registered in the international prospective register of systematic reviews (PROSPERO) with ID: CRD42022357027 guided by the Preferred Reporting Items for Systematic Reviews and Meta-Analyses (PRISMA) statement [13] and the Cochrane Handbook of Systematic reviews and meta-analysis [14].

### 2.2. Data Sources & Search Strategy

Web of Science, PubMed (MEDLINE), SCOPUS, EMBASE, and Cochrane Central Register of Controlled Trials (CENTRAL) were systematically searched by two reviewers (B.A. and M.A.) until 18 August 2022. We used no search limits or filters. The detailed search strategy and results are outlined in (Table 1).

### 2.3. Eligibility Criteria

We included RCTs with the following PICO criteria: population (P): adult patients who underwent successful TAVR and are without any indication of OAC use; intervention (I): DOACs; control (C): standard of care, including vitamin K antagonists, DAPT, or SAPT; outcomes (O): primary outcomes: mortality (all-cause mortality, mortality due to cardiovascular causes, and mortality due to non-cardiovascular causes). Secondary outcomes include the composite of mortality, myocardial infarction (MI), and stroke/transient ischemic attack (TIA); RLM (defined as at least one leaflet with grade ≥3 reduced motion); HALT (defined as at least one leaflet with hypoattenuated leaflet thickening); and major or life-threatening bleeding. The exclusion criteria involved studies only including patients with an indication of OAC such as AF, animal studies, cohort, retrospective, case series, case reports, non-randomized trials, laboratory studies, and conference abstracts.

### 2.4. Study Selection

After duplicates were removed using the Covidence online tool [15], two investigators (B.K. and M.G.) independently checked the eligibility of titles and abstracts of the imported records. Then, the full texts of the potentially relevant articles were checked according to the previously stated eligibility criteria. Any discrepancies were solved by inviting another investigator (B.A.) to reach a consensus.

### 2.5. Data Extraction

Using a pilot-tested extraction sheet, two reviewers (B.K. and M.G) separately extracted the following data from the included RCTs: trial characteristics (year of publication, country, study design, total participants, intervention regimen, control regimen, main inclusion criteria, and follow-up duration); baseline information (age, sex, number of patients in each group, body mass index (BMI), Society of Thoracic Surgery (STS) score, and comorbidities); and efficacy outcomes data (mortality, MI, stroke, major bleeding, life-threatening bleeding, HALT, and RLM). Disagreements were resolved by another investigator (B.A.) to reach a consensus.

### 2.6. Risk of Bias and Quality Assessment

Two reviewers (B.K. and M.G.) individually assessed the quality of the included RCTs, guided by The Cochrane Collaboration’s revised tool for assessing the risk of bias in randomized trials (ROB2) [16]. The following domains were considered: randomization process, deviations from intended interventions, missing outcome data, measurement of the outcome, and selection of the reported. Any discrepancy was resolved by inviting a third reviewer (M.A.) to reach a consensus. Two reviewers (B.A. and M.A.), guided by the Grading of Recommendations Assessment, Development, and Evaluation (GRADE) guidelines [17], appraised the quality of the outcome findings. Imprecision, indirectness, inconsistency, publication bias, and risk of bias were considered. Our results about the quality of evidence were rationalized, clarified, and included for each outcome. Any discrepancies were handled through discussion.

### 2.7. Statistical Analysis

This meta-analysis was conducted using Revman software version 5.4 (https://training.cochrane.org/online-learning/core-softward/revman (accessed on 11 October 2022)) to pool dichotomous outcomes using risk ratio (RR) as an effect measure along with the corresponding 95% confidence interval (CI). Pooled analysis was conducted using the fixed-effects model. Heterogeneity was evaluated using the chi-square test and measured using the I-square test. The chi-square test was considered significant on an alpha level below 0.1, and heterogeneity was considered significant if I-square was >50%.

## 3. Results

### 3.1. Search Results and Study Selection

A total of 738 studies were identified from PubMed (122), Web of Science (117), Cochrane (46), Scopus (307), and Embase (146). After removing duplicates (289), 449 records underwent title and abstract screening from which 436 were identified as irrelevant. Leaving 13 records to be considered by full-text screening excluding eight of them. Finally, we included three RCTs and two substudies in qualitative and quantitative data synthesis. The PRISMA flow chart of the detailed selection process is demonstrated in (Figure 1).

### 3.2. Characteristics of Included Studies

Three RCTs were included (GALILEO, ATLANTIS, and ADAPT-TAVR) [6,7,10] with two substudies (GALILEO-4D and ATLANTIS-4D-CT) [11,12] reporting leaflet outcomes (HALT and RLM) for the corresponding main trials [6,7] with a total of 2922 patients; 1463 in the DOACs group and 1459 in the placebo group. Included RCTs were open-label multicenter trials using three different factor Xa inhibitors: rivaroxaban [6,12], edoxaban [10], and apixaban [7,11]. Included trials’ characteristics are presented in (Table 2). The mean age of the DOACs group and placebo group was 80.9 and 81.4 years, respectively. The baseline characteristics of the participants are presented in (Table 3).

### 3.3. Risk of Bias and Quality of Evidence

Four trials [6,7,11,12] showed an overall high risk of bias derived mainly from the high risk of performance bias, being open-label trials, while Park et al. [10] showed some concerns regarding the overall risk of bias derived from concerns about selection and performance biases. A detailed outline of the risk of bias evaluation is illustrated in (Figure 2). Using the GRADE system, the included outcomes yielded low-quality evidence. Details and explanations are clarified in (Table 4).

### 3.4. Primary Outcomes

#### 3.4.1. All-Cause Mortality

All-cause mortality was higher in the DOAC group versus the standard of care (RR: 1.68 with 95% CI [1.22, 2.30], *p* = 0.001) (low-quality evidence) (Figure 3, Table 4). The pooled studies were homogenous (*p* = 1.00, I^2^ = 0%).

#### 3.4.2. Mortality Due to Cardiovascular Causes

There was no difference between DAACs and standard of care regarding mortality due to cardiovascular causes (RR: 1.36 with 95% CI [0.92, 2.03], *p* = 0.13) (low-quality evidence) (Figure 3, Table 4). The pooled studies were homogenous (*p* = 0.51, I^2^ = 0%).

#### 3.4.3. Mortality Due to Non-Cardiovascular Causes

Mortality due to non-cardiovascular causes was increased in the DOAC group as compared to the standard of care (RR: 2.34 with 95% CI [1.36, 4.02], *p* = 0.002) (low-quality evidence) (Figure 3, Table 4). The pooled studies were homogenous (*p* = 0.27, I^2^ = 24%).

### 3.5. Secondary Outcomes

#### 3.5.1. The Composite of Mortality, MI, and Stroke/TIA

The data favored the standard of care (RR: 1.41 with 95% CI [1.13, 1.76], *p* = 0.002) (low-quality evidence) (Figure 4A, Table 4). the pooled studies were homogenous (*p* = 0.71, I^2^ = 0%).

#### 3.5.2. RLM

RLM was lower with DOACs (RR: 0.19 with 95% CI [0.09, 0.41], *p* = 0.0001) (low quality evidence) (Figure 4B, Table 4). the pooled studies were homogenous (*p* = 0.44, I^2^ = 0%).

#### 3.5.3. HALT

HALT was lower with DOACs (RR: 0.50 with 95% CI [0.36, 0.70], *p* = 0.0001) (low quality evidence) (Figure 4C, Table 4). the pooled studies were homogenous (*p* = 0.56, I^2^ = 0%).

#### 3.5.4. Major or Life-Threatening Bleeding

There was no difference between DAACs and standard of care regarding the incidence of major or life-threatening bleeding (RR: 1.18 with 95% CI [0.90, 1.55], *p* = 0.24) (low-quality evidence) (Figure 4D, Table 4). the pooled studies were homogenous (*p* = 0.28, I^2^ = 21%).

## 4. Discussion

This meta-analysis showed that DOACs were significantly associated with a higher incidence of all-cause mortality; mortality due to the non-cardiovascular cause; and the composite efficacy outcome of mortality, MI, and stroke/TIA. However, DOACs were significantly associated with a decreased risk of RLM and HALT. Additionally, there was no difference between DOACs and the standard of care regarding mortality due to cardiovascular causes.

Accordingly, the observed increased risk of all-cause mortality and the composite of mortality, MI, and stroke/TIA was mainly based on mortality due to non-cardiovascular causes [7], which is consistent with our findings. This can be attributed to the high-risk profile of patients undergoing TAVR. Addition-ally, Dangas et al., reported that the reason behind the higher mortality due to non-cardiovascular causes is still unclear [6].

TAVR is a relatively new technique that is undermined by several knowledge gaps, including the mechanism, timing of leaflet thrombosis, and its effects on the valvular dynamics or durability, along with the incidence of stroke or systemic thromboembolism and the degree of bleeding risk associated antithrombotic therapy after TAVR [18]. Furthermore, the optimal antithrombotic therapy after TAVR is still to be defined with multiple recent RCTs comparing DOACs with the standard of care [6,7,10,11,12]. Therefore, we evaluated the efficacy and safety of DOACs versus the standard of care in patients without an indication of OACs usage. 

Multiple factors can increase stroke incidence after TAVR. The TAVR technique involves valve implantation through a relatively large delivery catheter leading to in-creased contact with the arterial wall [19] and subsequently increased risk of atherosclerotic emboli, especially in patients with aortic arch atheroma which is navigated by the bioprosthesis during valve implantation [18]. Accordingly, the patient undergoing TAVR may be more prone to stroke/TIA events because aortic stenosis prevalence es-calates with elder age (>75 years) who typically present with other concomitant comorbidities [18]. Furthermore, thrombosis developing on the bioprosthesis due to tissue injury, non-physiological valvular dynamic alterations, artificial surface expo-sure, and embolization of delayed stent frame can increase the risk of long-term stroke incidence after TAVR [18,20]. Thus, antithrombotic therapy use after TAVR according to the proper indications is justified [18]. 

Current guidelines by the ACC/AHA for the management of valvular heart dis-eases recommended aspirin 75 to 100 mg a day (SAPT) for patients without any other indications of OAC [9] which is supported by the safety evidence derived from multi-ple RCTs [4,5,6,7]. It should be highlighted that none of these trials were truly designed to explore if SAPT is equally effective in preventing ischemia and/or valve-related events following successful TAVR, compared to DAPT [18]. Moreover, the efficacy of clopidogrel alone is yet to be investigated which may be more beneficial than aspirin [18].

The risk of developing MI after TAVR is generally low with an incidence between 0% to 2.8% after one month and 0.4% to 3.5% after one year in the pivotal trials with no significant discrepancy with different risk profiles [16,19,20,21,22,23,24,25,26,27,28]. This is similar to the incidence of MI with the use of DOACs [6,10]. Therefore, the implementation of procedures preventing the risk of coronary obstruction such as chimney stenting [28] or native aortic scallop intentional laceration [29] can ameliorate the risk of MI incidence [18]. Moreover, valve design, coronary access, and the position of the leaflet posts according to the natural or bioprosthetic commissures need to be carefully considered to furtherly decrease the risk of MI [18,29,30,31,32,33]. However, the risk of late MI following TAVR should also be emphasized because patients who had TAVR and developed acute coronary syndromes have certain characteristics such as problems with coronary reaccess and low use of invasive procedures; hence, carrying a poor prognosis [18,34,35]. If a patient is undergoing TAVR with a history of recent percutaneous coronary intervention, a mandated application of DAPT is required [18]. However, DAPT in this group should be as short as possible (one to six months), to mitigate the increased risk of bleeding [36].

There was no difference between DOACs and the standard of care in preventing major or life-threatening bleeding. The risk of bleeding after TAVR is correlated with the patient’s comorbidities, necessitating the judicious application of antithrombotic therapy [18]. Moreover, the risk of developing serious bleeding was higher than developing a major stroke, warranting a cautious approach, especially in high-risk patients [19]. This also highlights the challenges in treating TAVR patients who are usu-ally elderly with multiple coexisting comorbidities increasing the risk of both bleeding and thromboembolic events [6]. Several factors may be associated with an increased risk of bleeding after TAVR, including non-stopping OAC perioperatively, clopidogrel loading, and presentation with atrial fibrillation [18]. Therefore, the adoption of other stroke-preventing measures such as cerebral embolic protection devices to prevent perioperative stroke can be applied to mitigate the risk of bleeding by decreasing the dosage or the duration of the required antithrombotic therapy [18]. Moreover, post-procedural bleeding beyond 30 days was not adequately defined and tracked in most TAVR trials, which may furtherly necessitate investigating the intensity and duration of antithrombotic therapy, especially in elder patients [18]. 

Despite that the relationship between RLM and HALT with thromboembolic events is still a matter of debate [23,37,38], some may argue that it is associated with more incidence of stroke/TIA [39,40,41]. In our analysis, DOACs were significantly associated with decreased rates of RLM and HALT; however, this was not reflected in the prevention of stroke/TIA, new cerebral lesions on MRI, or new neurocognitive dysfunction with similar rates, compared to standard of care [10,26]. Therefore, it should be highlighted that subclinical leaflet thrombosis has no effect on clinical outcomes in patients who underwent TAVR and it should not determine the optimal antithrombotic therapy [10]. In the same line, this may not support the adoption of routine imaging screening tests for detecting subclinical leaflet thrombosis and imaging-guided antithrombotic therapy [10].

A research letter discussing DOACs after TAVR was recently published during the preparation and reviewing process of our manuscript, which may limit the novelty of our manuscript [42]. However, the methods of this review meticulously followed the PRISMA guidelines [13] and the Cochrane Handbook of Systematic reviews and me-ta-analysis [14], performing a thorough statistical analysis, assessing the risk of bias using Cochrane ROB2 tool [13], and evaluating the quality of evidence using GRADE working group recommendations [17].

### 4.1. Limitations

Our review has limitations. First, there were only three included RCTs with a small sample size limiting the generalizability of our findings [6,7,10]. Second, the DOACs and the standard of care regimens (Table 2) varied among the included trials which may affect our findings; however, we detected minimal to no heterogeneity in our meta-analysis. Third, the results of HALT and RLM are based on a single screening session reported to be a dynamic process [43,44]. Fourth, there was no screening after ischemic events or after stopping antithrombotic therapy; hence, a clear association between subclinical leaflet thrombosis and thromboembolic events could not be clarified. Fifth, none of the included RCTs reported the time of incidence of stroke or hemorrhagic events. Finally, HALT may be associated with leaflet thrombosis; however, only ADAPT-TAVR reported the incidence of leaflet thrombosis [10] and none of the included trials included pathological confirmation [6,7,10].

### 4.2. Implications for Future Research

Future trials are required to address the evidence gap on the safety and efficacy of SAPT using clopidogrel [18]. More research is required on the optimal strategy in managing TAVR patients, on the role of antiplatelet versus antithrombotic therapy in preventing HALI and RLM, and on the role of computed tomography in the screening for subclinical leaflet thrombosis [18,19].

## 5. Conclusions

DOACs significantly prevented HALT and RLM; however, it was associated with poorer efficacy and safety outcomes, compared to the standard of care. Therefore, more large-scale RCTs are still required to investigate the optimal antithrombotic regimen after TAVR.

## Figures and Tables

**Figure 1 jcm-11-06781-f001:**
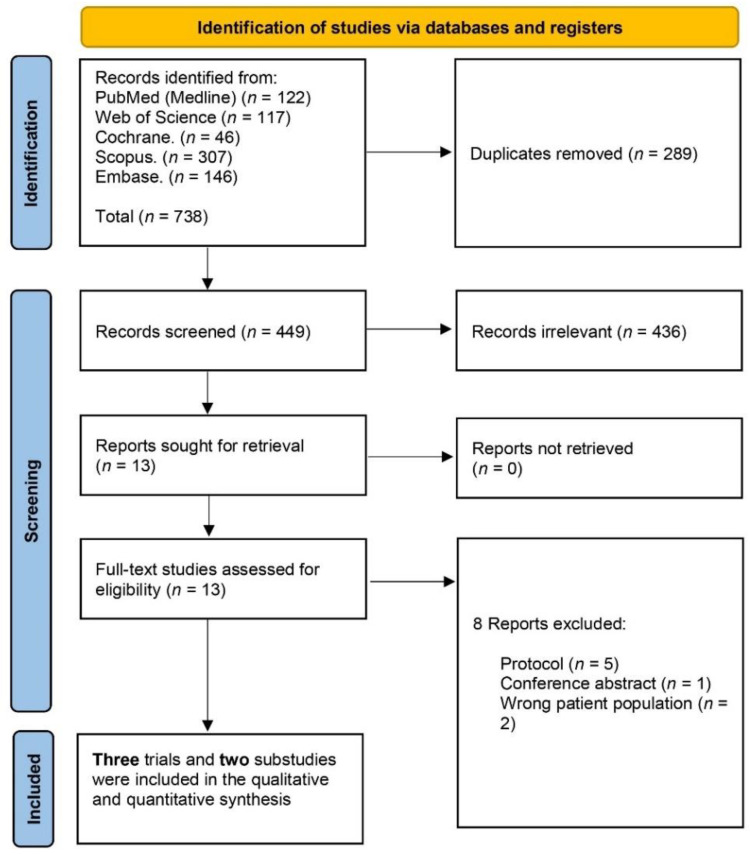
PRISMA flow chart of the screening process.

**Figure 2 jcm-11-06781-f002:**
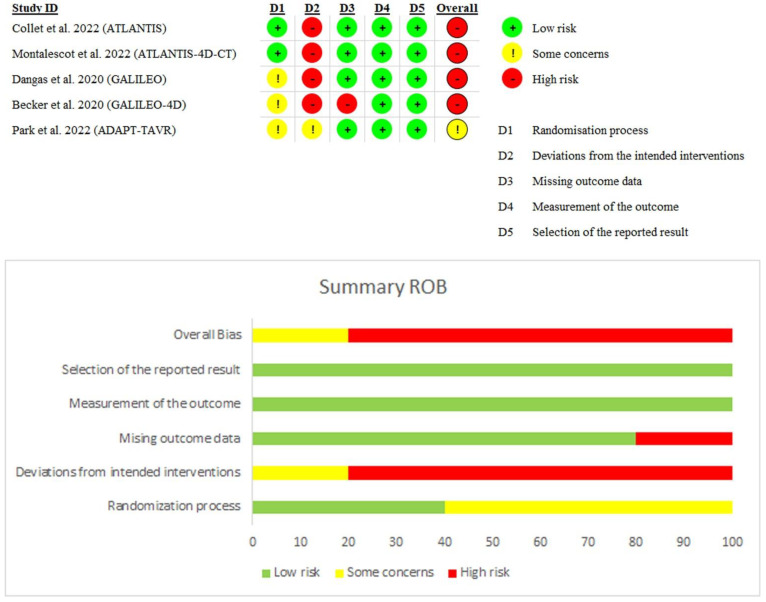
Quality assessment of risk of bias in the studies. The upper panel [5,6,7,8,9] presents a schematic representation of risks (low = green, unclear = yellow, and high = red) for specific types of biases of each of the studies. The lower panel presents risks (low = green, unclear = yellow, and high = red) for the subtypes of biases of the combination of studies included in this review.

**Figure 3 jcm-11-06781-f003:**
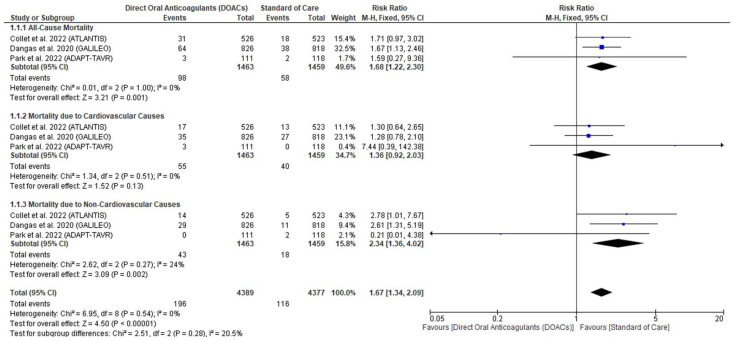
Forest plot of the primary outcome (Mortality). RR: risk ratio, CI: confidence interval [5,7,8].

**Figure 4 jcm-11-06781-f004:**
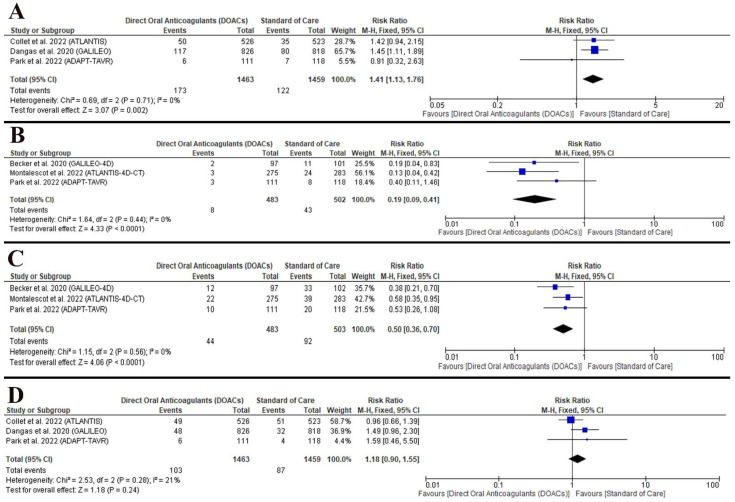
Forest plot of the secondary outcomes ((**A**) the composite mortality, MI, and stroke/TIA; (**B**) RLM; (**C**) HALT; (**D**) major or life-threatening bleeding). RR: risk ratio, CI: confidence interval [5,6,7,8,9].

**Table 1 jcm-11-06781-t001:** Search terms and results in different databases.

Database	Search Terms	Search Field	Search Results
PubMed	(“direct oral anticoagulant*” OR DOAC* OR NOAC* OR apixaban OR rivaroxaban OR edoxaban) AND (“transcatheter aortic valve” OR TAVR OR TAVI)	All Field	122
Cochrane	(“direct oral anticoagulant*” OR DOAC* OR NOAC* OR apixaban OR rivaroxaban OR edoxaban) AND (“transcatheter aortic valve” OR TAVR OR TAVI)	All Field	46
WOS	(“direct oral anticoagulant*” OR DOAC* OR NOAC* OR apixaban OR rivaroxaban OR edoxaban) AND (“transcatheter aortic valve” OR TAVR OR TAVI)	All Field	117
SCOPUS	TITLE-ABS-KEY ((“direct oral anticoagulant*” OR doac* OR noac* OR apixaban OR rivaroxaban OR edoxaban) AND (“transcatheter aortic valve” OR tavr OR tavi))	Title, Abstract	307
EMBASE	#3. #1 AND #2 #2. ‘transcatheter aortic valve’: ti,ab,kw OR tavr:ti,ab,kw OR tavi:ti,ab,kw #1. ‘direct oral anticoagulant’: ti,ab,kw OR doac:ti,ab,kw OR noac:ti,ab,kw OR apixaban:ti,ab,kw OR rivaroxaban:ti,ab,kw OR edoxaban:ti,ab,kw	All Field	146

**Table 2 jcm-11-06781-t002:** Characteristics of the included studies.

Study ID	Study Design	Total Participants	DOACs			Standard of Care			Follow-Up (Months)	Primary Outcome
			Drug	Dose	Dosing	Drug	Dose	Dosing		
Collet et al., 2022 (ATLANTIS) [8]	Open-label, multicenter RCT	1049	Apixaban	5 or 2.5 mg	Twice a day	Vitamin K antagonist, SAPT, or DAPT	According to the physician’s decision	N/A	12	The composite of all-cause mortality, MI, stroke/TIA, SE, intracardiac or valve thrombosis, DVT/PE, and life-threatening or major bleeding
Montalescot et al., 2022 (ATLANTIS-4D-CT) [6]	Open-label, multicenter RCT	558	Apixaban	5 or 2.5 mg	Twice a day	Vitamin K antagonist, SAPT, or DAPT	According to the physician’s decision	N/A	12	At least 1 prosthetic valve leaflet with RLM of grade 3 or 4 or HALT of grade 3 or 4
Dangas et al., 2020 (GALILEO) [7]	Open-label, multicenter RCT	1644	Rivaroxaban plus Aspirin	Rivaroxaban (10 mg) plus Aspirin (75 to 100 mg)	Once daily	DAPT	Aspirin (75 to 100 mg) plus clopidogrel (75 mg)	Once daily	24	The composite of all-cause mortality, MI, stroke/TIA, SE, valve thrombosis, and DVT/PE
Becker et al., 2020 (GALILEO-4D) [9]	Open-label, multicenter RCT	231	Rivaroxaban plus Aspirin	Rivaroxaban (10 mg) plus Aspirin (75 to 100 mg)	Once daily	DAPT	Aspirin (75 to 100 mg) plus clopidogrel (75 mg)	Once daily	3	At least 1 prosthetic valve leaflet with RLM of grade 3 or 4 or HALT of grade 3 or 4
Park et al., 2022 (ADAPT-TAVR) [5]	Open-label, multicenter RCT	229	Edoxaban	60 or 30 mg	Once daily	DAPT	Aspirin (100 mg) plus clopidogrel (75 mg)	Once daily	6	Valve leaflet thrombosis

RCT: randomized controlled trial, DOACs: direct oral anticoagulants, DAPT: dual antiplatelet therapy, SAPT: single antiplatelet therapy, N/A: not available, mg: milligram, MI: myocardial infarction, TIA: transient ischemic attack, SE: systemic embolism, DVT: deep venous thrombosis, PE: pulmonary embolism, RLM: reduced leaflet motion, HALT: hypoattenuated leaflet thickening.

**Table 3 jcm-11-06781-t003:** Baseline characteristics of the participants.

Study ID	Number of Patients		Age (Years), Mean (SD)		Gender (Male), N (%)		BMI, Mean (SD)		STS Score, Mean (SD)		Comorbidities, N. (%)											
											HTN		HF		DM		CAD		Stroke or ICH		Permanent Pacemaker	
	DOACs	Control	DOACs	Control	DOACs	Control	DOACs	Control	DOACs	Control	DOACs	Control	DOACs	Control	DOACs	Control	DOACs	Control	DOACs	Control	DOACs	Control
Collet et al., 2022 (ATLANTIS) [8]	749	751	81.6 (6.1)	82.3 (6.4)	344 (45.9)	360 (47.9)	27.52 (5.45)	27.33 (5.16)	5.14 (5.02)	5.14 (5.38)	606 (80.9)	601 (80)	292 (39.0)	284 (37.8)	221 (29.5)	214 (28.5)	N/A	N/A	78 (10.4)	89 (11.9)	N/A	N/A
Montalescot et al., 2022 (ATLANTIS-4D-CT) [6]	275	283	81.5 (6.1)	82.3 (6.3)	170 (45.9)	180 (45.9)	27.4 (5.4)	27.1 (4.7)	4.9 (4.2)	4.9 (5.8)	294 (79.5)	310 (79.1)	139 (37.6)	144 (36.7)	110 (29.7)	114 (29.1)	N/A	N/A	40 (10.8)	51 (13.0)	N/A	N/A
Dangas et al., 2020 (GALILEO) [7]	826	818	80.4 (7.1)	80.8 (6.0)	426 (51.6)	405 (49.5)	28.1 (5.5)	28.2 (5.7)	4.0 (3.2)	4.3 (3.5)	720 (87.2)	697 (85.2)	394 (47.7)	380 (46.5)	236 (28.6)	235 (28.7)	325 (39.3)	305 (37.3)	51 (6.2)	35 (4.3)	80 (9.7)	80 (9.8)
Becker et al., 2020 (GALILEO-4D) [9]	115	116	79.7 (7.3)	80.5 (6.2)	74 (64.3)	74 (63.8)	27.7 (6.5)	27.8 (5.1)	2.8 (1.5)	3.0 (2.1)	98 (85.2)	95 (81.9)	52 (45.2)	52 (44.8)	21 (18.3)	27 (23.3)	42 (36.5)	36 (31.0)	11 (9.6)	6 (5.2)	14 (12.2)	14 (12.1)
Park et al., 2022 (ADAPT-TAVR) [5]	111	118	80.2 (5.2)	80 (5.3)	49 (44.1)	47 (39.8)	24.8 (3.8)	24.8 (4.3)	3.1 (2.1)	3.5 (2.7)	81 (73.0)	84 (71.2)	17 (15.3)	12 (10.2)	35 (31.5)	36 (30.5)	32 (28.8)	34 (28.8)	N/A	N/A	N/A	N/A

DOACs: direct oral anticoagulants, N/A: not available, BMI: basal metabolic index, HTN: hypertension, HF: heart failure, DM: diabetes mellitus, CAD: coronary artery disease, ICH: intracranial hemorrhage, STS: Society of Thoracic Surgeon, SD: standard deviation, N: number.

**Table 4 jcm-11-06781-t004:** GRADE evidence profile.

Certainty Assessment							No of Patients		Effect		Certainty	Importance
Number of Studies	Study Design	Risk of Bias	Inconsistency	Indirectness	Imprecision	Other Considerations	Intervention	Comparison	Relative (95% CI)	Absolute (95% CI)		
**All-Cause Mortality**												
**3**	randomized trials	serious	not serious	not serious	serious ^b^	none	98/1463 (6.7%)	58/1459 (4.0%)	RR 1.68 (1.22 to 2.30)	27 more per 1000 (from 9 more to 52 more)	⨁⨁◯◯ Low	CRITICAL
**Mortality due to Cardiovascular Causes**												
**3**	randomized trials	serious ^a^	not serious	not serious	serious ^b^	none	55/1463 (3.8%)	40/1459 (2.7%)	RR 1.36 (0.92 to 2.03)	10 more per 1000 (from 2 fewer to 28 more)	⨁⨁◯◯ Low	CRITICAL
**Mortality due to Non-Cardiovascular Causes**												
**3**	randomized trials	serious ^a^	not serious	not serious	serious ^c^	none	43/1463 (2.9%)	18/1459 (1.2%)	RR 2.34 (1.36 to 4.02)	17 more per 1000 (from 4 more to 37 more)	⨁⨁◯◯ Low	CRITICAL
**The composite of mortality, MI, and stroke/TIA**												
**3**	randomized trials	serious ^a^	not serious	not serious	serious ^b^	none	173/1463 (11.8%)	122/1459 (8.4%)	RR 1.41 (1.13 to 1.76)	34 more per 1000 (from 11 more to 64 more)	⨁⨁◯◯ Low	IMPORTANT
**RLM**												
**3**	randomized trials	serious ^a^	not serious	not serious	serious ^c^	none	8/483 (1.7%)	43/502 (8.6%)	RR 0.19 (0.09 to 0.41)	69 fewer per 1000 (from 78 fewer to 51 fewer)	⨁⨁◯◯ Low	CRITICAL
**HALT**												
**3**	randomized trials	serious ^a^	not serious	not serious	serious ^c^	none	44/483 (9.1%)	92/503 (18.3%)	RR 0.50 (0.36 to 0.70)	91 fewer per 1000 (from 117 fewer to 55 fewer)	⨁⨁◯◯ Low	CRITICAL
**Major or Life-threatening bleeding**												
**3**	randomized trials	serious ^a^	not serious	not serious	serious ^c^	none	103/1463 (7.0%)	87/1459 (6.0%)	RR 1.18 (0.90 to 1.55)	11 more per 1000 (from 6 fewer to 33 more)	⨁⨁◯◯ Low	IMPORTANT

CI: confidence interval, RR: risk ratio, MI: myocardial infarction, TIA: transient ischemic attack, RLM: reduced leaflet motion, HALT: hypoattenuated leaflet thickening, ^a^: Included RCTs show an overall high risk of bias, ^b^: The confidence interval (CI) of the pooled risk ratio does not exclude the risk of appreciable harm/benefits, ^c^: The pooled analysis included less than 300 events.

## Data Availability

All data are included in the manuscript.

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
