# Peer review of "The Efficacy and Safety of Direct Oral Anticoagulants versus Standard of Care in Patients without an Indication of Anti-Coagulants after Transcatheter Aortic Valve Replacement: A Meta-Analysis of Randomized Controlled Trials"

_jcm, 2022, doi:10.3390/jcm11226781_

Round 1
Reviewer 1 Report
The authors presented a meta-amalysis studying the efficacy and safety of DOAC in patients without indication of OAC and who recievd TAVR.
3 RCT were included including 2922 patients divided in 2 groups (DOACs vs Placebo).
They concluded that DOACs were significantly associated with a higher incidence of all-cause mortality, mortality due to the non-cardiovascular cause, and the composite efficacy outcome of mortality, MI, and stroke/TIA. However, DOACs were significantly associated with a decreased risk of reduced leaflet motion (RLM) and hypoattenuated leaflet thickening (HALT). Additionally, there was no difference between DOACs and the standard of care regarding mortality due to cardiovascular causes.
The study is well written, the statistical analysis is appropriate.
Author Response
We sincerely thank reviewer 1 for their comments, we really appreciate their time and consideration to improve the manuscript.
Reviewer 2 Report
The chosen studies were not constructed to evaluate HALT and RLM therefore data on those parameters must be secondary and not systematic observations.Could you find also data on the time of stroke or haemorragic events after TAVI implantation? Different f.i. between clop+aspir and DOACs?
Author Response
The chosen studies were not constructed to evaluate HALT and RLM therefore data on those parameters must be secondary and not systematic observations.Could you find also data on the time of stroke or hemorrhagic events after TAVI implantation? Different f.i. between clop+aspir and DOACs?
Thank you so much for you for your feedback, we appreciate your efforts to improve our manuscript. Please see our response to each points below.
- The chosen studies were not constructed to evaluate HALT and RLM therefore data on those parameters must be secondary and not systematic observations.
Thank you for this comment, we agree with the reviewer that these results are secondary, and we highlighted this in lines 196-199 in 3.5.2. RLM. and in lines 200-203 in 3.5.3. HALT. in 3.5. Secondary Outcomes. Also, it has been mentioned in lines 303-304 in the 4.1 Limitations section as follows:
. . . the results of HALT and RLM are based on a single screening session reported to be a dynamic process.
- Could you also find data on the time of stroke or hemorrhagic events?
Thank you for pointing this out, unfortunately there is no reported data on time of stroke or hemorrhagic events after TAVR in the included studies, it has been added to the lines 307-308 in 4.1 Limitations to be considered in the future research as follows:
. . . none of the included RCTs reported the time of incidence of stroke or hemorrhagic events.
- Different f.i. between clop+aspir and DOACs?
Thank you for your comment, two of the included trials compared between DOACs and clop+aspir. Only the ATLANTIS trial compared DOACs and vitamin K antagonist, SAPT, or DAPT (patient tailored management), as illustrated in Table 2. We have discussed the potential of DAPT in the management of patients without and indication for anticoagulants after TAVR in lines 239 - 246 in 4. Discussion as follows:
Current guidelines by the American College of Cardiology/American Heart Association for the management of valvular heart diseases recommended aspirin 75 to 100 mg a day (SAPT) for patients without any other indications of OAC [19] which is supported by the safety evidence derived from multiple RCTs [7,8,20,21]. It should be highlighted that none of these trials were truly designed to explore if SAPT is equally effective in preventing ischemia and/or valve-related events following successful TAVR, compared to DAPT [16]. Moreover, the efficacy of clopidogrel alone is yet to be investigated which may be more beneficial than aspirin [16].

Round 2
Reviewer 2 Report
the paper was appropriately revised.
Author Response
We thank Reviewer 2 for their valuable comments.